# Tailored cobalt-salen complexes enable electrocatalytic intramolecular allylic C–H functionalizations

Chen-Yan Cai[1], Zheng-Jian Wu[1], Ji-Ying Liu[2], Ming Chen[1], Jinshuai Song [2] & Hai-Chao Xu [1✉]

Oxidative allylic C–H functionalization is a powerful tool to streamline organic synthesis as it minimizes the need for functional group activation and generates alkenyl-substituted products amenable to further chemical modifications. The intramolecular variants can be used to construct functionalized ring structures but remain limited in scope and by their frequent requirement for noble metal catalysts and stoichiometric chemical oxidants. Here we report an oxidant-free, electrocatalytic approach to achieve intramolecular oxidative allylic C–H amination and alkylation by employing tailored cobalt-salen complexes as catalysts. These reactions proceed through a radical mechanism and display broad tolerance of functional groups and alkene substitution patterns, allowing efficient coupling of di-, tri- and even tetrasubstituted alkenes with N- and C-nucleophiles to furnish high-value heterocyclic and carbocyclic structures.

[1] Key Laboratory of Chemical Biology of Fujian Province, State Key Laboratory of Physical Chemistry of Solid Surfaces, and College of Chemistry and Chemical Engineering, Xiamen University, Xiamen, China. [2] Green Catalysis Center, College of Chemistry, Zhengzhou University, Zhengzhou, China. ✉email: haichao.xu@xmu.edu.cn

atalytic allylic C–H functionalizations are particularly attractive transformations because they maximize step- and atom-economy and the retained or newly formed alkenyl moiety is available for versatile further transformations[1–3]. Intramolecular variants of these transformations can provide rapid access to functionalized hetero- and carbocycles. These allylic C–H functionalization reactions are generally catalyzed by noble metals such as Pd, Ru, and Ir and involve either the formation of an allyl-metal complex via C–H activation followed by it being attacked by a tethered nucleophile, or sequential nucleometallation and β-hydride elimination (Fig. 1a)[1–3]. A noticeable drawback of the allyl-metal pathway is that its application to an internal alkene substrate usually produces a mixture of regioisomers, limiting most reactions to terminal alkenes[3,4]. The nucleometallation pathway, on the other hand, is compatible with internal alkenes and has been extensively studied since 1970s for the synthesis of N-heterocycles[5–9]. These intramolecular oxidative amination reactions, referred to as Aza-Wacker cyclizations, can be achieved under mild conditions using palladium catalysis with oxygen as the terminal oxidant, although the cyclization of densely substituted alkenes remains challenging[10–15].

Radical cyclizations are broadly compatible with various alkene substitution patterns because of the high reactivities of the open shell species[16,17]. However, radical-mediated catalytic allylic C–H functionalizations with nucleophiles remain rare due to the lack of catalysts for oxidative radical formation and termination[18]. Nonetheless, since the first report in 1980s, there has been a plethora of studies on intramolecular allylic C–H alkylation of 1,3-dicarbonyl compounds with stoichiometric Mn(III) and Cu(II) salts, which serve the purposes of oxidizing the substrate to a C-radical and converting the cyclized alkyl radical to an alkene, respectively (Fig. 1b)[19,20]. Following a similar mechanism, we have accomplished Cu(II)-catalyzed intramolecular allylic C–H amination with DMP as the terminal oxidant (Fig. 1c)[21]. The use of stoichiometric sacrificial oxidants poses a significant challenge to product isolation and reaction scale-up[22]. In addition, 3° alkyl radicals are easily oxidized to carbocations by the Cu(II) salts, resulting in by-product formation and yield reduction[23].

An ideal approach to dehydrogenative transformations is through H2 evolution without using any chemical oxidants[24,25], which can be conveniently achieved via organic electrochemistry[26–39]. We[40,41] and others[42] have reported electrochemical allylic C–H amination

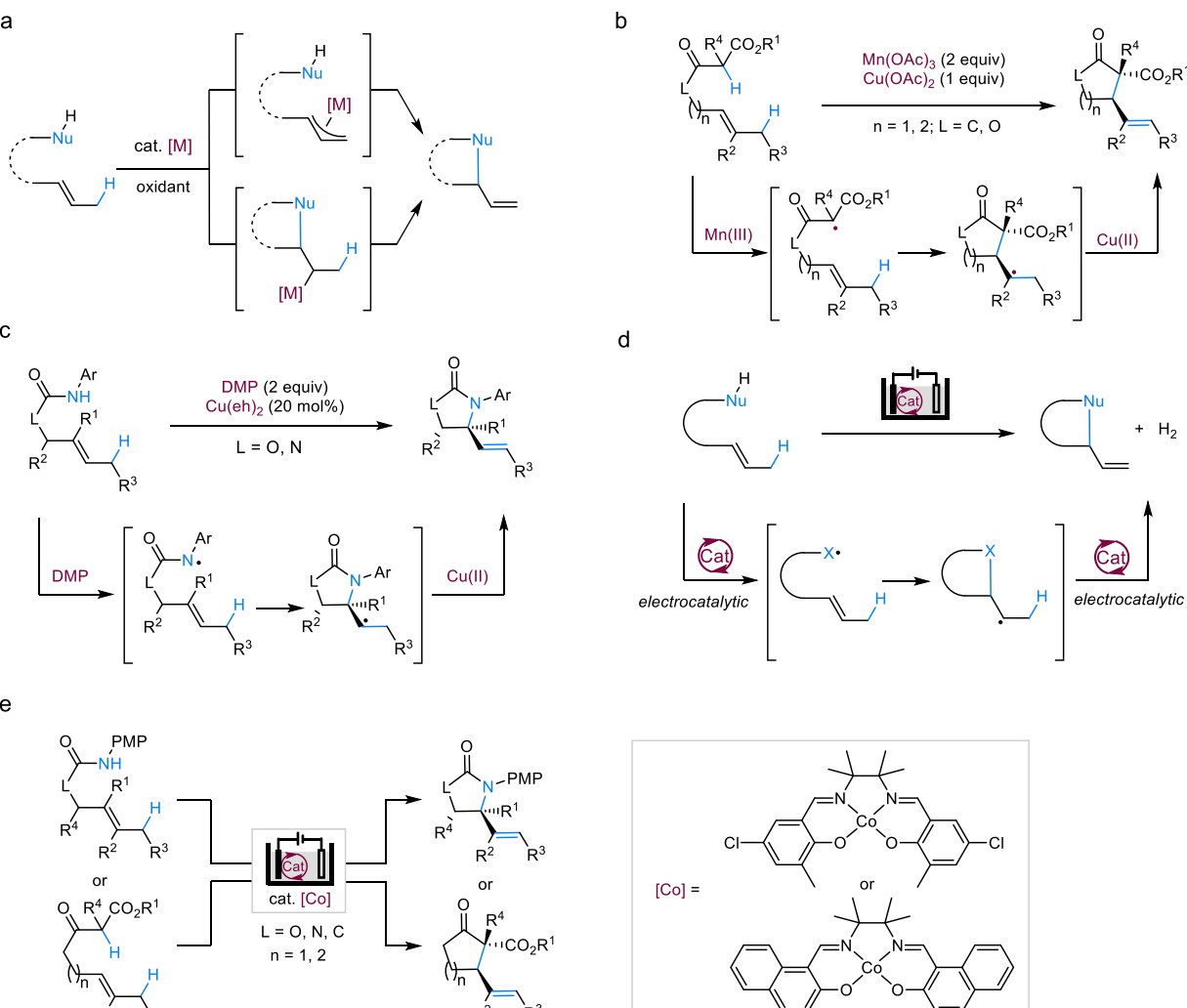

**Fig. 1 Intramolecular oxidative allylic C–H functionalization. a** Transition-metal catalyzed allylic C–H functionalization usually proceeds through C–H activation to form allyl-metal species or nucleometallation/β-hydride elimination. **b** Mn(III)/Cu(II)-mediated intramolecular allylic C–H alkylation of 1,3-dicarbonyl compounds. **c** DMP-promoted, Cu(II)-catalyzed intramolecular allylic C–H amination. **d** Proposed electrocatalytic approach to achieve intramolecular allylic C–H functionalization. The electrocatalyst facilitates both the initial substrate oxidation and the conversion of the cyclized alkyl radical to an alkene. **e** Electrocatalytic intramolecular allylic C–H amination and alkylation enabled by tailored cobalt-salen complexes. DMP Dess–Martin periodinane, Cu(eh)2 copper(II) 2-ethylhexanoate.

through direct electrolysis or Cu(II)-mediation, which is highly useful but far from perfect. On the one hand, direct electrolysis is often incompatible with disubstituted internal alkenes[40]; on the other hand, electrolysis in the presence of Cu(II) requires inconvenient divided cells to avoid Cu(II) reduction and a large excess of supporting salts to increase conductivity[22,42]. In addition, both approaches rely on direct oxidation of substrates on the anode, which necessitates a high anodic potential and consequently limits the scope of functional groups allowed. Furthermore, there is no efficient electrochemical methods for the allylic alkylation reaction. Building on our experience in the electrocatalytic generation of N- and C-centered radicals[17,43–45], we argue that the aforementioned challenges can be addressed by electrocatalytic allylic C–H functionalization (Fig. 1d).

In this work, we report an efficient and broadly applicable electrocatalytic strategy for the intramolecular allylic C–H amination and alkylation in undivided cells, by employing tailored cobalt-salen complexes as catalysts, which are low in oxidation potentials and well suited for converting alkyl radicals to alkenes (Fig. 1e).

## Results and discussion

**Reaction development.** The electrocatalytic cyclization of carbamate **1** bearing a removable N-PMP group[46] was chosen as a model reaction for optimization (Table 1 and Supplementary Table 1). We chose to screen a wide variety of cobalt(II) salen complexes as catalysts because they are known to oxidize alkyl radicals to alkenes[47,48], and share similar oxidation potentials as ferrocene[49], which had been previously employed to catalyze the electrooxidative generation of N- and C-centered radicals[43,44]. The electrolysis was conducted under a constant current in a three-necked round-bottomed flask equipped with a reticulated vitreous carbon (RVC) anode and a platinum plate cathode. The optimal conditions comprised cobalt-salen complex **[Co]-1** as a catalyst, $Na_2CO_3$ as a basic additive, and $Et_4NPF_6$ as supporting

electrolyte, in a mixed solvent of MeCN/MeOH (5:1). Under these conditions, the desired product **2** was obtained in 86% yield as the only observed diastereomer and regioisomer (Table 1, entry 1). Both the Co-catalyst (entry 2) and heating (entry 3) were critical for product formation, whereas omitting $Na_2CO_3$ hampered the conversion of **1** and thus decreased the yield of **2** to 52% (entry 4). Reduction of the amount of **[Co]-1** to 5 mol% did not affect the yield of **2** but led to the formation of a small amount of side product **2'** (4% yield) that was inseparable from **2** (entry 5). While $K_2CO_3$ was similarly effective as $Na_2CO_3$, other bases such as NaOPiv, $NaHCO_3$, $K_2HPO_4$, and LiOMe were less efficient in promoting the formation of **2** (Supplementary Table 1). The use of $Cs_2CO_3$ as a base resulted in the decomposition of the carbamate **1** to generate MeOCONHPMP in 60% yield and no formation of **2** (Supplementary Table 1). The amount of $Et_4NPF_6$ could be reduced to 0.1 equiv. without affecting the reaction efficiency (entry 6). Replacing **[Co]-1** with another Co-catalyst that differs in electronic and steric properties, such as **[Co]-2**, **[Co]-3**, or **[Co]-4**, led to sub-optimal results (entry 7). We also tested several other cathode materials and found brass (entry 8) and nickel (entry 9) to be equally suitable as platinum, but not graphite (entry 10), probably because of its high overpotential for proton reduction. Pleasingly, the electrolysis could be performed under air without any apparent yield loss (entry 11).

**Evaluation of substrate scope.** We next explored the reaction scope by varying the alkenyl moiety and the nitrogen nucleophile (Fig. 2). N-PMP carbamates tethered to a di- (**3–15**), tri- (**16–23**) or even tetrasubstituted (**24–27**) alkene readily underwent allylic C–H amination, but the less electron-rich N-Ph carbamate (**4a**) was found to be much less reactive[21]. In addition to carbamates, the reaction was broadly compatible with N-PMP amides (**28–32**) and ureas (**33–41**), and could also be applied on other nucleophiles, such as N-alkoxyamides (**42–44**), sulfonyl hydrazines (**45**,

---

**Table 1 Optimization of reaction conditions for intramolecular allylic C–H amination.**

**[Co]-1**, R$^1$ = Cl, R$^2$ = Me
**[Co]-2**, R$^1$ = CF$_3$, R$^2$ = tBu
**[Co]-3**, R$^1$ = OMe, R$^2$ = tBu
**[Co]-4**, R$^1$ = tBu, R$^2$ = tBu

| Entry | Deviation from standard conditions | Yield of 2 (%)[a] |
|---|---|---|
| 1 | None | 86[b] |
| 2 | No **[Co]-1** | 0 |
| 3 | Reaction at rt | Trace |
| 4 | No $Na_2CO_3$ | 52 (46)[c] |
| 5 | 5 mol% of **[Co]-1** | 87[d] |
| 6 | $Et_4NPF_6$ (0.1 equiv.) | 85 |
| 7 | **[Co]-2**, or **[Co]-3**, or **[Co]-4** as catalyst | 50–63 |
| 8 | Brass (1 cm × 1 cm) as cathode | 82 |
| 9 | Nickel plate (1 cm × 1 cm) as cathode | 83 |
| 10 | Graphite plate (1 cm × 1 cm) as cathode | 29 (58)[c] |
| 11 | Under air | 86 |

Reaction conditions: RVC (1 cm × 1 cm × 1.2 cm), Pt plate cathode (1 cm × 1 cm), **1** (0.2 mmol), MeCN (5 mL), MeOH (1 mL), constant current = 10 mA, 4 h (7.5 F mol$^{-1}$). Compound **2** was formed as a single diastereomer under all conditions.
*PMP* p-methoxyphenyl.
[a]Determined by $^1$H NMR analysis using 1,3,5-trimethoxybenzene as the internal standard.
[b]Isolated yield.
[c]Unreacted **1** in brackets.
[d]**2'** was formed in a 4% yield.

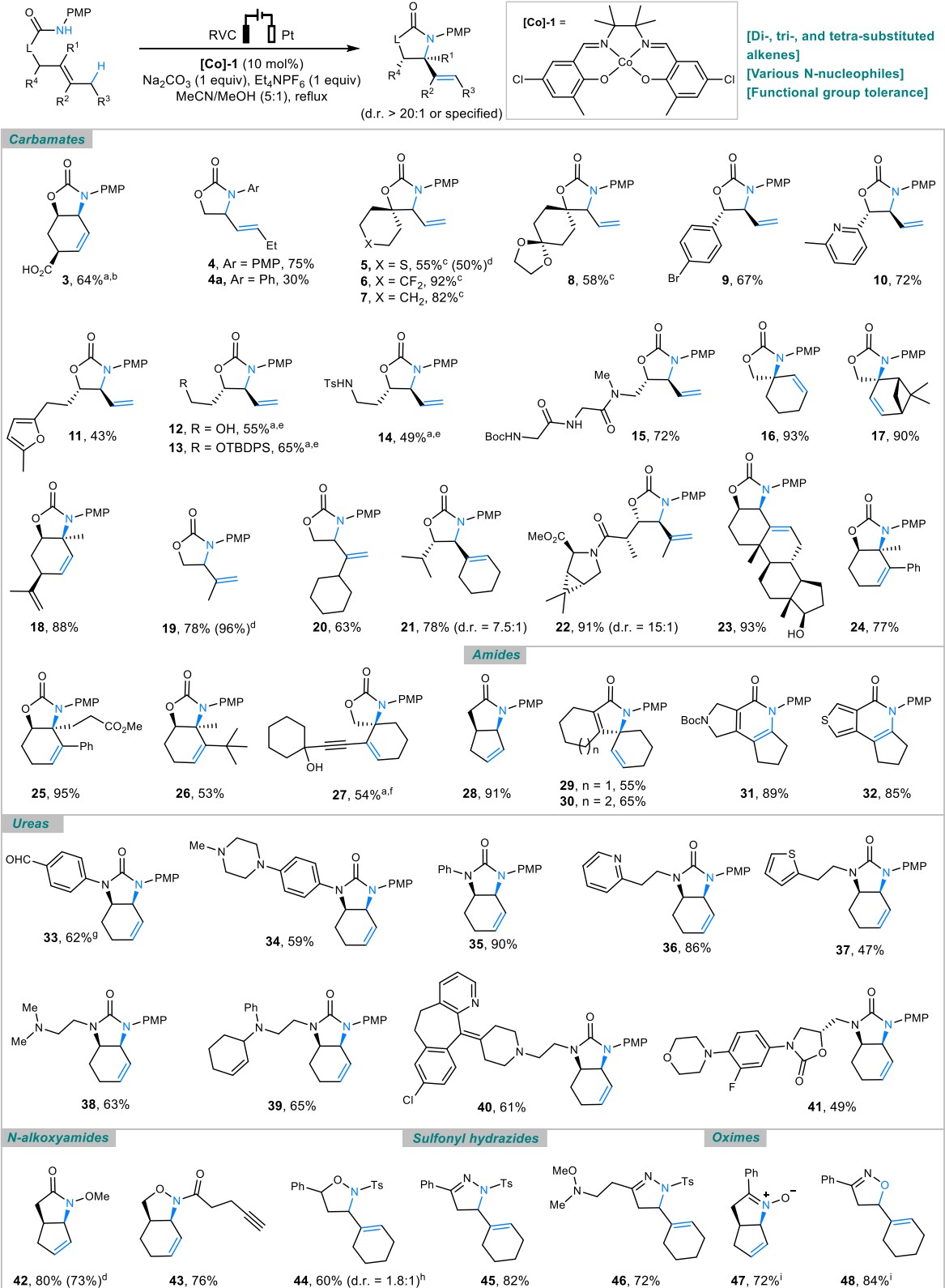

**Fig. 2 Scope of allylic C–H amination.** Reaction conditions: substrate (0.2 mmol), MeCN (5 mL), MeOH (1 mL), 1.5–7.0 h (2.8–13.0 F mol⁻¹). All yields are isolated yields. [a]Reaction with 15 mol% of **[Co]-1**. [b]Reaction with 1.5 equiv. of Na₂CO₃. [c]Current = 8 mA. [d]Reaction with 0.1 equiv. of Et₄NPF₆. [e]Current = 7 mA. [f]Current = 6 mA. [g]Reaction at 60 °C. [h]Reaction at rt. [i]K₂CO₃ as base instead of Na₂CO₃. d.r. diastereomeric ratio, Ts tosyl, Boc tert-butyloxy carbonyl.

**46**) and oximes (**47**, **48**), to generate various alkene-functionalized *N*-heterocycles. In general, acceptable nucleophilic substrates share the chemical feature of having an acidic N–H or O–H bond, and their conjugate bases are low in oxidation potential. It should be emphasized that all products were derived from *exo*-type cyclization except **31** and **32**, in which cases the fused five-membered *N*- or *S*-heterocyclic ring forced the reaction to choose the 6-endo pathway[50]. The electrocatalytic method tolerated a wide range of functional groups, including those that are base/acid-sensitive, such as chiral carboxylic acid (**3**), ketal (**8**), *N*-Boc amines (**15**, **31**), and amino ester (**22**), as well as the ones susceptible to oxidation, such as thioether (**5**), furan (**11**), alcohols (**12**, **23**, **27**), sulfonamide (**14**), aldehyde (**33**), and amines (**34**, **38**–**41**, **46**). For the reactions of the oximes, the use of K$_2$CO$_3$ instead of Na$_2$CO$_3$ as a base was found to be important to obtain

synthetically useful yields. Although the reactions were conducted with 1 equiv. of Et$_4$NPF$_6$, it was possible to reduce its amount to 0.1 equiv. as demonstrated with the synthesis of compounds **5**, **19**, and **42**.

We next investigated the allylic C–H alkylation reactions of 1,3-dicarbonyl compounds such as malonate ester (**49**), β-ketoesters (**50**–**72**), and malonate amides (**73**–**78**) (Fig. 3a). These alkylation reactions required different conditions for optimal results. After reaction optimization (Supplementary Table 2), we determined that a different cobalt-salen complex, [**Co**]-**5**, was necessary to achieve the best results. Pleasingly, most of the intramolecular alkylation reactions proceeded efficiently at rt. probably because of the relatively low oxidation potentials of the 1,3-dicarbonyl-derived carbanions. The formation of 5-membered carbocyclic products was substantially less diastereoselective than that of 6-

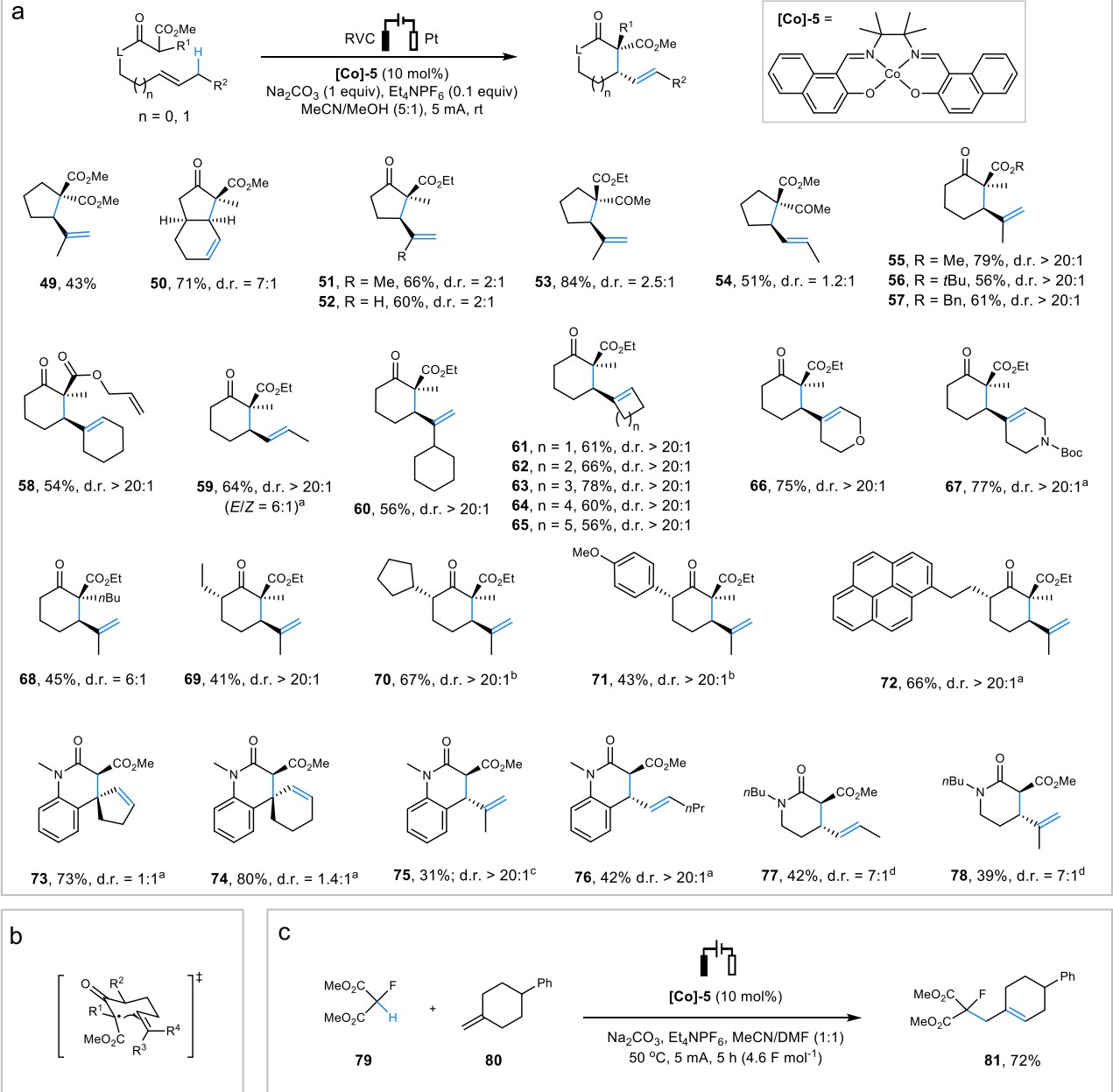

**Fig. 3 Scope of allylic C–H alkylation. a** Intramolecular allylic C–H alkylation. Reaction conditions: substrate (0.3 mmol), MeCN (5 mL), MeOH (1 mL), rt, 10 mA, 3.8–4.6 h (2.4–2.8 F mol$^{-1}$). **b** Proposed transition state for 6-*exo* cyclizations. **c** Intermolecular allylic C–H alkylation. [a]Reaction in reflux glyme (5 mL)/MeOH (1 mL) with *n*Bu$_4$NOAc (1 equiv.) as electrolyte. [b]Reaction in the presence of 1 equiv. of Et$_4$NPF$_6$. [c]Reaction under reflux. [d]Reaction in the presence of NaOAc (1 equiv.) in reflux MeCN (6 mL)/H$_2$O (0.3 mL) at 2.5 mA for 8.0 h. Bn benzyl.

membered cyclic ketones, including those bearing three stereo-centers (e.g., **69–72**). The excellent stereoselectivity of the 6-*exo* cyclization reactions could be attributed to kinetic control and the involvement of a chairlike transition state (Fig. 3b and Supplementary Fig. 1)[23,51]. Our method also proved efficient in converting malonate amides, which are less acidic than malonate ester and β-ketoester, to δ-lactams (**75–78**). All reactions except those of trisubstituted cyclic alkenes (**73**, **74**) exhibited satisfactory stereoselectivity. In contrast, no successful Mn(III)/Cu(II)-mediated cyclization of malonate amides has been reported. Intermolecular allylic C–H alkylation was also possible, as demonstrated by the reaction of malonate **79** with alkene **80** in MeCN/DMF (Fig. 3c).

We next applied our electrochemical method on a decagram scale to further demonstrate its utility. With the help of larger electrodes to allow for higher electric current, we successfully synthesized 10.3 g of **2** in 71% yield, and 12.2 g of **63** in 88% yield (Fig. 4a), the latter of which even improved over what we accomplished on the milligram scale (78%). The scalability of our method coupled with versatile known transformations of the alkene-bearing cyclic carbamate and urea products to valuable structures such as vicinal amino alcohols and diamines makes the electrosynthesis attractive for synthetic applications[40,46,52]. Additional merits of our approach included mitigated sensitivity to steric hindrance and enhanced selectivity in the H-elimination step compared to the use of Cu(II) salts. For example, while the Co-catalyzed electrochemical reaction of trisubstituted alkene **83** afforded the intended allylic amination product **18**, a similar but Cu(II)-catalyzed reaction of the analogous **84** ended up in cascade cyclization to give **85** (Fig. 4b). These results suggested that the Co-catalyst that we used was more efficient than Cu(II) in catalyzing the oxidation of sterically hindered 2° alkyl radical

intermediates to alkenes. The alkylation of 1,2-disubstituted alkene **86** produced a regioisomeric mixture of **54** and **87** with the use of stoichiometric Mn(III)/Cu(II) salts[23], but furnished **54** exclusively under electrocatalytic conditions (Fig. 4c). Additionally, Mn(III)/Cu(II)-mediated reaction of **88** formed undesired **89** due to overoxidation of the 3° alkyl radical intermediate to a carbocation and its subsequent solvolysis (Fig. 4d)[23]; in contrast, this was not observed with Co-catalysis.

**Mechanistic investigation.** As part of the mechanistic studies, the cyclopropane-bearing substrate **90** was cyclized under the standard reaction conditions to furnish a pair of ring-opening products **91** and **92**, suggesting the involvement of an alkyl radical **93** (Fig. 5a)[53]. Reaction of **1** with 3 equiv. of preformed $[Co^{III}]^+$ **95** in the presence of $Cs_2CO_3$ under reflux afforded **2** in 20% yield with 78% unreacted **1** (Fig. 5b). Meanwhile, no reaction occurred at rt. or when an insoluble base such as $Na_2CO_3$ was employed. These results suggested that **95** could oxidize **1** to **2** but required heating and a base. Cyclic voltammetry (Fig. 5c) revealed that the electrocatalyst **[Co]-1** (red trace) could undergo reversible single-electron oxidation ($E_{p/2} = 0.31$ V) to $[Co^{III}]^+$ or reduction ($E_{p/2} = -1.25$ V) to $[Co^I]^-$ in MeCN/MeOH (5:1). The low oxidation potential of **[Co]-1** underpinned the broad functional group tolerance of the electrocatalytic method. While the voltammogram of $[Co^{II}]/[Co^{III}]^+$ in the presence of **1** remained largely unchanged regardless of whether $Na_2CO_3$ was added (without $Na_2CO_3$, cyan trace; with $Na_2CO_3$, olive trace), the employment of NaOMe as a substitute base caused the curve to shift negatively and the reduction wave to disappear, but no catalytic current was observed (black trace). Combined, we deduced that the electrochemically generated $[Co^{III}]^+$ reacted with the conjugate base of **1** but without reverting to **[Co]-1** at rt[54]. Interestingly,

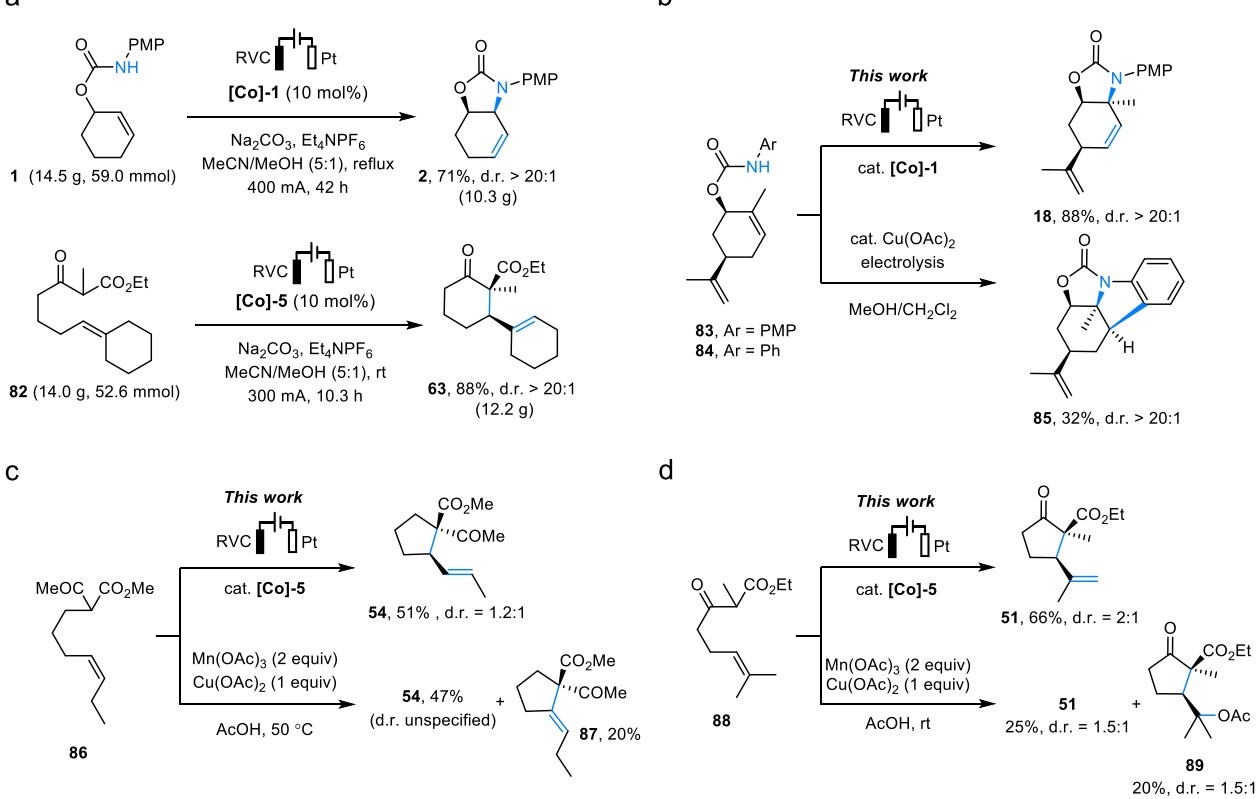

**Fig. 4 Decagram scale synthesis and comparison of Co-catalyzed versus Cu-based allylic C–H functionalizations. a** Decagram scale synthesis of **2** and **63**. **b** Comparison of the reaction patterns of Co- versus Cu-catalyzed cyclization of sterically hindered alkenes. **c**, **d** Comparison of selectivities of Co-catalyzed versus Mn/Cu-mediated allylic C–H alkylation of di- and trisubstituted alkenes.

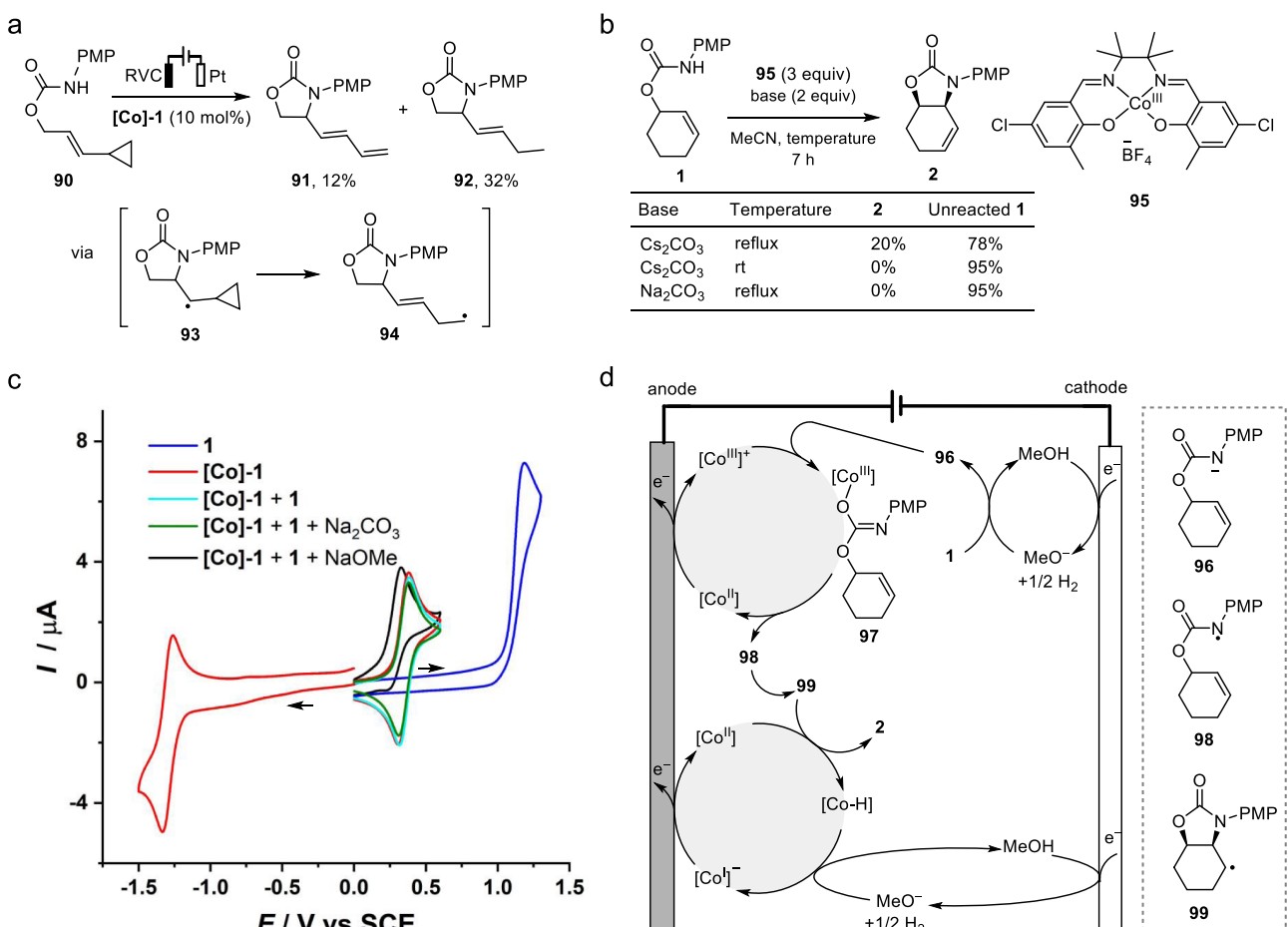

**Fig. 5 Mechanistic studies and proposal. a** Reaction of radical clock substrate **90**. **b** Reaction of **1** with stoichiometric Co$^{III}$ complex **95**. **c** Cyclic voltammograms obtained in 0.1 M Et$_4$NPF$_6$, MeCN/MeOH (5:1). The red (positive part), cyan and olive traces overlapped, suggesting that there was no reaction between [Co$^{III}$]$^+$ and **1** under these conditions. **d** Proposed reaction mechanism for Co-catalyzed intramolecular allylic C–H amination.

preparative electrolysis of **1** without using an alkaline additive afforded **2** in 52% yield (see Table 1, entry 4). We speculated that MeO$^-$ generated at the cathode might have served as a base to promote substrate oxidation[43,45].

Based on the above information, we formulated a possible mechanism for our electrocatalytic allylic C–H amination reaction. The Co-catalyst [Co$^{II}$] is first oxidized to [Co$^{III}$]$^+$ at the anode[55], and MeOH undergoes cathodic reduction to generate MeO$^-$ and H$_2$. The carbamate **1** is then deprotonated by MeO$^-$ and the resultant conjugate base **96** coordinates with the [Co$^{III}$]$^+$ complex to form **97**, which then affords amidyl radical **98** and regenerates the [Co$^{II}$] catalyst via heat-induced homolytic dissociation. Subsequent cyclization of **98** furnishes alkyl radical **99**, which is oxidized by [Co$^{II}$] to the amination product **2** and a [Co–H] species via either direct H transfer or addition/β-hydride elimination[47,48]. The intermediate C-radical **99** may be reduced with [Co–H] species or the solvent to generate side product **2′**. Finally, deprotonation of the [Co–H] species by MeO$^-$ leads to a [Co$^I$]$^-$ complex[48,56] that is oxidized back to [Co$^{II}$] at the anode. The allylic C–H alkylation likely follows a similar mechanism.

In summary, we have developed an electrocatalytic approach for intramolecular allylic C($sp^3$)–H amination and alkylation. These reactions are catalyzed by molecular cobalt-salen complexes, which oxidize the substrate to initiate radical cyclization and convert the cyclized alkyl radical to the alkene product. The favorable redox potential of the cobalt catalyst, coupled with its

high reactivity for the oxidation of 2° and 3° alkyl radical intermediate, contributes to broad reaction compatibility with a diverse range of functional groups and alkene substitution patterns, thereby enabling the efficient synthesis of various high-value alkene-bearing cyclic structures. The synthetic utility of our method is further enhanced by its scalability and elimination of external chemical oxidants. We are convinced that the use of tunable cobalt-salen complexes as molecular electrocatalysts for the oxidative formation and termination of radicals will create vast opportunities for reaction discovery.

## Methods

**Representative procedure for allylic C–H amination**. The substrate (0.2 mmol, 1 equiv.), Et$_4$NPF$_6$ (0.2 mmol, 1 equiv.), Na$_2$CO$_3$ (0.2 mmol, 1 equiv.), and [**Co**]-**1** (0.02 mmol, 0.1 equiv.) were placed in a 10 mL three-necked round-bottom flask. The flask was equipped with a condenser, a reticulated vitreous carbon (RVC) anode (100 PPI, 1 cm × 1 cm x 1.2 cm), and a platinum plate (1 cm × 1 cm × 0.1 cm) cathode (Supplementary Fig. 3a). The flask was flushed with argon. MeCN (5 mL) and MeOH (1 mL) were added. The electrolysis was carried out at reflux using a constant current of 10 mA until complete consumption of the substrate (monitored by TLC or $^1$H NMR). The reaction mixture was cooled to rt. H$_2$O (30 mL) and EtOAc (30 ml) were added. The phases were separated, and the aqueous phase was extracted twice with EtOAc. The combined organic solution was dried over anhydrous MgSO$_4$, filtered, and concentrated under reduced pressure. The residue was chromatographed through silica gel eluting with EtOAc/hexanes to give the product. All new compounds were fully characterized (see the Supplementary Methods).

**Representative procedure for allylic C–H alkylation**. A 10 mL three-necked round-bottom flask was charged with [**Co**]-**5** (0.03 mmol, 10 mol%), Et$_4$NPF$_6$

(0.03 mmol, 0.1 equiv.) and $Na_2CO_3$ (0.3 mmol, 1 equiv.). The flask was equipped with a reticulated vitreous carbon anode (100 PPI, 1 cm × 1 cm × 1.2 cm) and a platinum plate (1 cm × 1 cm) cathode. The flask was flushed with argon. The substrate (0.3 mmol, 1 equiv.), MeCN (5 mL) and MeOH (1 mL) were added. The electrolysis was carried out at rt. or reflux using a constant current of 5 mA until complete consumption of the substrate (monitored by TLC or $^1$H NMR). The reaction mixture was concentrated under reduced pressure. The residue was chromatographed through silica gel eluting with EtOAc/hexanes to give the desired product. All new compounds were fully characterized (see the Supplementary Methods).

## Data availability

The data supporting the findings of this study are provided within the article and in the Supplementary Information file. Any further relevant data are available from the authors on request.

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

## Acknowledgements

We acknowledge the financial support of this research from NSFC (No. 21971213), MOST (2016YFA0204100), and Fundamental Research Funds for the Central Universities. J.S. thanks the Henan Province Supercomputing Center. We thank Dr. Yan Liu (XMU) for help with mass spectroscopy.

## Author contributions

C.-Y.C. and Z.-J.W. contributed equally. C.-Y.C., Z.-J.W., and M.C. performed the experiments and analyzed the data. J.-Y.L and J.S. conducted DFT calculations. H.-C.X. designed and directed the project and wrote the manuscript.

## Competing interests

The authors declare no competing interests.
