## [Peer Review File · Nature Communications]

REVIEWER COMMENTS

Reviewer #1 (Remarks to the Author):

The study by Xu and co-workers investigates the intramolecular cyclisation (allylic cyclisation) mediated by a cobalt(II)salen complex. The reaction is proposed to proceed through a radical cyclisation mechanism catalysed by cobalt, which is oxidised electrochemically.

The reaction reported is based on the groups previously reported electrochemically induced radical cyclisation of N-centred radicals (for example summarised in Acc Chem Res 2019 52 3339)

There seem to be three critical problems with the optimisation studies, and the so-called standard conditions (Table 1, entry 1).

The optimisation studies found that 0.1 equivalents of supporting electrolyte was sufficient to promote the reaction with the best yields obtained in the table (entry 5). In addition, if the reaction could be conducted in the presence of air (entry 10).

The yield was maintained with a cobalt loading of 5 mol% (Supplementary Table 1).

Given this, it seems that some combination of these factors (lower cobalt and electrolyte loading, and in the presence of air) should be used as the optimised conditions for the scoping studies, and not those given in entry 1.

The optimisation studies do not report any control reactions – these should be carried out.

The optimisation studies reported in Table 1 should include diastereomeric ratios for all entries, or a footnote that the product was formed as a single diastereomer in all cases.

The authors have tested a variety of substrates which is commendable.

Figure 2 should include diastereomeric ratios for all relevant products.

Authors should comment on the change of base that was required for products in Figure 2 with the footnote h. Why was this required?

Authors should comment on the difference in the optimised conditions required for amination and alkylation. What was the difference in the two cobalt complexes, and why was this required?

Mechanistic studies

A reference should be provided for the radical ring-opening reaction reported in Figure 5a.

It is not clear to me why the reactions reported in Figure 5b are not conducted under the standard conditions. These reactions should be repeated in order to draw any conclusions relating to the mechanism.

Why did the reaction require 3 equivalents of Co(III) species? Reaction with 1 equivalent only should be sufficient.

Can the electrochemical studies be repeated under heated conditions in order to see if a catalytic current could be observed?

What literature is available to support the proposal that methanol is involved in the reaction?

What is the outcome of conducting the reaction using deuterated solvent? A series of reactions should be carried out: deuterated acetonitrile, deuterated methanol, both deuterated solvent.

Can the authors show that the carbamate can be deprotonated by MeO-?

Can the authors isolate or otherwise characterise any of the cobalt intermediate proposed in their catalytic cycle (for example [Co-H] or 97)?

The experimental procedures appear to be well described.

Computational studies were completed using an inappropriate basis set. These studies should be benchmarked on either literature methods or experimental values. Coordinates for all calculated structures should be provided, or the reference to the database where they are deposited included in the supporting information.

The ¹H NMR spectra of numerous products showed that there was contamination. These products should be further purified, and the spectra re-taken. This is exemplified in products 14, 15 and 25.

Zoom ins of relevant ^1H and ^{13}C NMR splitting patterns should be added. For example, the ^1H NMR region from 5.5 – 6.5 ppm of product 3.

Reviewer #2 (Remarks to the Author):

Hai-Chao Xu and coworkers have done a splendid job describing an efficient and novel means of achieving intramolecular allylic C–H functionalization. The chemistry occurs electrocatalytically using tailored cobalt-salen complexes as catalysts. The authors do an excellent job placing their work within the context of prior art and provide convincing arguments detailing the improvements their work brings to the synthesis toolbox. The substrate scope is large and the yields are excellent. The chemistry can be scaled to decagram levels and occurs without the use of external chemical oxidants. The mechanistic scheme, illustrated in Figure 5, is sound – it resting upon the results of a set of well crafted and insight providing control experiments.

I believe the research is of broad interest and will surely attract the attention of researchers in both academics and industry. The chemistry has the potential for widespread use.

I recommend publication “as is”, except for one suggestion. Thus, I believe that it would behoove the authors to include a few explicit comments regarding the manipulations that can be carried out on both the urea and the oxazolidinone containing adducts. I believe this would enhance the interest.

Reviewer #3 (Remarks to the Author):

Recommendation: Publish after minor revisions noted

A cobalt-salen complexes catalyzed intramolecular cyclizations enabled by electrocatalysis with a broad tolerance of functional groups and alkene substitution patterns in was described this submit. This method could be applied on a decagram scale synthesis. Control experiments and cyclic voltammetry experiments were carried out to prove the mechanism. However, it is necessary to make some revises before publication:

1 Pay attention to the statement, such as the statements at Line 18 and 21; Line 144, [CoI]- should be [CoI]-

2 Could it achieve enantioselectivity if using chiral Salen cobalt catalysis?

3 These literatures should be cited: “Acc. Chem. Res. 2019, 52, 3309–3324”; “CCS Chem. 2020, 2233-2244”; “Chin. J. Chem. 2021, 39, 143-148”.

A point-to-point response to the comments of reviewers is as following. The original reviewer comments are in black and our responses are in blue.

Reviewer #1 (Remarks to the Author):

The study by Xu and co-workers investigates the intramolecular cyclisation (allylic cyclisation) mediated by a cobalt(II)salen complex. The reaction is proposed to proceed through an radical cyclisation mechanism catalysed by cobalt, which is oxidised electrochemically.

The reaction reported is based on the groups previously reported electrochemically induced radical cyclisation of N-centred radicals (for example summarised in Acc Chem Res 2019 52 3339)

Response: We thank the reviewer for taking time to evaluate the manuscript and give suggestions to improve it.

There seem to be three critical problems with the optimisation studies, and the so-called standard conditions (Table 1, entry 1).

The optimisation studies found that 0.1 equivalents of supporting electrolyte was sufficient to promote the reaction with the best yields obtained in the table (entry 5). In addition, if the reaction could be conducted in the presence of air (entry 10). The yield was maintained with a cobalt loading of 5 mol% (Supplementary Table 1). Given this, it seems that some combination of these factors (lower cobalt and electrolyte loading, and in the presence of air) should be used as the optimised conditions for the scoping studies, and not those given in entry 1.

Response: The reviewer has made a good point. The reasons that we did not do that are as following.

1. Catalyst loading. The target product was formed in good NMR yield of 87% with 5 mol% of catalyst loading for the model reaction of carbamate **1**. But under these conditions, the reaction also afforded 4% yield of hydroamidation product, compound **2'** of Supplementary Table 1 (also see below). The side product **2'** is inseparable from the target product **2**. Hence, we did not use 5 mol% catalyst for scope studies. 5 mol% loading may be employed in cases that the hydroamidation product can be removed (e.g. via recrystallization) or if it is not a problem for further applications.

2. Supporting electrolyte. The manuscript contained two reactions, one amination and one alkylation reaction. The story is that we finished the studies of the amination reaction first and then decided to expand to alkylation. We try to publish less and decide to combine all the results in one paper. During the studies of the amination reactions, we used 1 equiv of supporting electrolyte and did not consider lowering the concentration of electrolyte. It was during the investigations of the alkylation we decided to reduce the use of salts and found that it was possible to use just 0.1 equiv of electrolyte. Hence, most of reactions of alkylation (Fig 3) were conducted with 0.1 equiv of Et₄NPF₆, including the decagram scale reaction (Fig 4a, bottom). Since we have already finished the scope studies of the amination, we did not try to repeat all the experiments with 0.1 equiv of supporting electrolyte. We simply showed in the optimization table that it was possible to reduce the use of salt for the amination if needed. But note that the reduction in the use of salt led to an increase of the cell potential [$E(\text{cell}) = 2.4 \text{ V}$ with 1 equiv of Et₄NPF₆ vs $E(\text{cell}) = 8.5 \text{ V}$ with 0.1 equiv of Et₄NPF₆] because of the increase in solution resistance. We have found that in some cases that the use of 1 equiv of salt was beneficial for the cyclization reactions such as the formation of carbocycles **70** and **71**.

To address the reviewer's comment, we have briefly studied the aminations with 0.1 equiv of Et₄NPF₆ under otherwise the standard conditions for the synthesis of compounds **5**, **19** and **42**. The results have been added to Fig 2 (also see below). While all the reactions were successful, the reduction in use of supporting electrolyte resulted in higher yield for **19**, slightly lower yields for **5** and **42**. Overall, it is possible to reduce the use of supporting electrolyte for the amination reaction if needed. It remains unknown how the salt concentration affects the yield. The change of salt concentration will change the thickness of electric double layer and affect the redox reactions of ionic species.

5	19	42	
Yield with 1.0 equiv of Et ₄ NPF ₆	55% (1.0 equiv)	78% (1.0 equiv)	80% (1.0 equiv)
Yield with 0.1 equiv of Et ₄ NPF ₆	50% (0.1 equiv)	96% (0.1 equiv)	73% (0.1 equiv)

3. Air. We conducted the model reaction under air to show that there is no need for stringent removal of oxygen from the reaction system. But running the reactions under air cause safety issues because the limiting oxygen concentration of the solvents such as MeCN (12.7%) and MeOH (8.6%) is relatively low (*Org. Process Res. Dev.* 2015, 19, 1537–1543). Hence, flammable atmosphere is formed with these solvents under air, which creates safety hazard. We thus conducted all the reactions under inert atmosphere.

The optimisation studies do not report any control reactions – these should be carried out.

Response: For the amination reaction, we have shown in Table 1 and Supplementary Table 1 the control experiments for the catalyst, heating, basic additive, and electricity. For the alkylation reaction, we have shown in Supplementary Table S2 control

experiments for catalyst, basic additive, and electricity. We are not sure what control reactions the reviewer is referring to.

The optimisation studies reported in Table 1 should include diastereomeric ratios for all entries, or a footnote that the product was formed as a single diastereomer in all cases.

Response: The product was formed as a single diastereomer under all conditions. We have added this information to the Table footnote.

The authors have tested a variety of substrates which is commendable.

Figure 2 should include diastereomeric ratios for all relevant products.

Response: we have stated under the title reaction in Figure 2 “d.r. >20:1 or specified”. The ratio has been specified if diastereomers are formed.

Authors should comment on the change of base that was required for products in Figure 2 with the footnote h. Why was this required?

Response: The reactions of the oximes with Na₂CO₃ as the base produced low yields. For example, **48** was formed in 40% yield with Na₂CO₃ as base (vs 84% with K₂CO₃). We have added the following comment to the text: “For the reactions of the oximes, the use of K₂CO₃ instead of Na₂CO₃ as a base was found to be important to obtain synthetically useful yields.”.

Although these basic additives are barely soluble under the conditions, their presence do increase the reaction efficiency (see Table 1). Of course, some carbonate may have dissolved during the reaction. The reaction may also take place on the surface of the carbonate. We are also interested in figuring out why sometimes we have to change the base because we have similar observations in many other reactions (e.g. *Chin. J. Chem.* **2018**, 36, 909). Oximes and anilides have different pKa and oxidation potentials and are expected to react differently. But it remains unclear to us how these properties are related to the use of difference bases. We cannot explain why the subtle differences of the barely soluble bases sometimes have dramatic effect on the reaction efficiency.

Authors should comment on the difference in the optimised conditions required for amination and alkylation. What was the difference in the two cobalt complexes, and why was this required?

Response: We have stated in the text “These alkylation reactions required different conditions for optimal results. After reaction optimization (Supplementary Table 2), we determined that a different cobalt-salen complex, **[Co]-5**, was necessary to achieve the best results. Pleasingly, most of the intramolecular alkylation reactions proceeded efficiently at rt probably because of the relatively low oxidation potentials of the 1,3-dicarbonyl-derived carbanions.”

The reaction conditions were screened for both reactions with model substrates carbamate **1** and β -ketoester **S130** (Table 1, Supplementary Tables 1 and 2). And the conditions were chosen based on the results of the model reactions. We cannot give a clear explanation on the change of each of the parameters. Here are our thoughts.

The differences of the two conditions include the cobalt catalyst, reaction temperature, current density, and concentration of substrate and supporting electrolyte. These two reactions involve the oxidative generation and reaction of different radical species. In general, both electronic and steric properties of the catalyst affect the reaction efficiency. Although the two reactions do need different cobalt complexes for optimal results, catalyst **[Co]-1** works for both amination and alkylation. The alkylation reaction of **S130** did afford the alkylation product **50** in 62% of compound with catalyst **[Co]-1**, slightly lower than the optimal 72% with catalyst **[Co]-5** (Supplementary Table 2). Hence, we employed **[Co]-5** as the catalyst for the alkylation reaction. Both Co-complexes are oxidized reversibly on the time scale of the CV (100 mV/s) with close oxidation potentials (0.31 V for **[Co]-1** and 0.30 V for **[Co]-5**). At this stage, it is difficult to explain why **[Co]-5** worked slightly better for the alkylation, especially considering the moderate difference. The substrates for amination and alkylation are different in their acidities [pKa (DMSO), about 12-15 for β -ketoesters and 1,3-diesters, about 20 for anilide] and are oxidized at different potentials (e.g. $E_{p/2} = 0.32$ V vs SCE for the conjugate base of **1**; $E_{p/2} = 0.14$ V for the conjugate base of β -ketoester **S130**, see CV in the SI). These differences mean that they react with different thermodynamics and kinetics. The lower oxidation potential of the carbon anion may explain the lower reaction temperature for the alkylation reaction. Note that although the conjugate bases of the substrates are oxidized at close or lower potential than the cobalt catalyst, the substrates are not oxidized directly on the electrode because there is not enough concentration of the substrate anions in solution to sustain the current. It is also important to note that the addition of stoichiometric soluble and strong bases to deprotonate the substrates causes substrate and catalyst decomposition.

Each reaction required an optimal current density. For a successful catalytic reaction, the oxidized catalyst needs to react with the substrate before it gets reduced at the cathode. Hence, the current density is screened to match the turnover of the catalyst. The cobalt catalyst probably go through Co(II)-Co(III)-Co(I) states. If current density is too high, there will probably not enough Co(II) to convert the intermediate C-radical to the final alkene. If the current density is too low, then [Co-H] species formed are not removed efficiently, which may cause side reactions such as reaction with the alkene moiety of the product or substrate (*J. Am. Chem. Soc.* **2014**, *136*, 16788; *J. Am. Chem. Soc.* **2019**, *141*, 9548).

We have explained the salt concentration above. As for the concentration of the starting material, it can be either 0.050 M or 0.033 M for both reactions. These two reactions were run at different concentrations simply because of the preference of the students performing the experiments. It is possible to run the reactions at higher concentration to reduce use of solvent and supporting electrolyte. For the decagram scale reaction of the alkylation, the concentration of starting material was 0.063 M.

Mechanistic studies

A reference should be provided for the radical ring-opening reaction reported in Figure 5a.

Response: The following reference on radical clock has been added as ref 53: Newcomb, M. Radical Kinetics and Clocks. *Encyclopedia of Radicals in Chemistry, Biology and Materials* (2012).

It is not clear to me why the reactions reported in Figure 5b are not conducted under the standard conditions. These reactions should be repeated in order to draw any conclusions relating to the mechanism.

Why did the reaction required 3 equivalents of Co(III) species? Reaction with 1 equivalent only should be sufficient.

Response: We have conducted additional experiments and included them in the SI as Supplementary Table 3 (also see below).

Supplementary Table 3 Addition experiments with stoichiometric [Co^{III}]^a

Entry	96	Base	Solvent	2	Unreacted 1
1	1 equiv	Na ₂ CO ₃ (1 equiv)	MeCN/MeOH (5:1)	0%	95%
2	2 equiv	Cs ₂ CO ₃ (1 equiv)	MeCN/MeOH (5:1)	0%	90%
3	1 equiv	Cs ₂ CO ₃ (1 equiv)	MeCN	8%	80%

^aReactions were conducted with 0.05 mmol of **1**.

1. As for the stoichiometric of Co(III), two equiv is needed for the two-electron oxidation reaction because the final state of Co species should be Co(I). Co(I) is highly reducing [$E_{p/2}(\text{Co}^{\text{III}}) = -1.25 \text{ V vs SCE}$] and will react with Co(III) to form Co(II).

2. Note that it is difficult to mimic the electrosynthesis with stoichiometric oxidants. For the reaction to occur with stoichiometric oxidants, a stoichiometric base is needed to deprotonate the substrate and remove [Co-H] species. [Co-H] specie can add to alkenes and needs to be removed efficiently. The challenge is that Co(III) gets reduced in the presence of stoichiometric bases, especially in hot and basic MeOH solution. The reaction of **1** with 1 equiv of Co(III) under “standard conditions” [e.g. MeOH/MeCN (1:5) as solvent, Na₂CO₃ (1 equiv) as additive, reflux] led to no reaction of **1** and quick decomposition of Co(III) to Co(II) as judged by the change of the color from dark green [color for Co(III)] to red [color for Co(II)] (Supplementary Table 3, entry 1). Under these conditions, the substrate was not deprotonated and thus did not react. Increasing the amount of Co(III) to 2 equiv and the use of Cs₂CO₃ (1 equiv) as the base to deprotonate the substrate also led to no formation of **2** with the recovery of most **1** (90%) because of the quick decomposition of the Co(III) in MeOH solution (Supplementary Table 3, entry 2). The oxidation of **1** with

1 equiv of Co(III) in MeCN in the presence of Cs₂CO₃ (1 equiv) resulted in the formation of **2** in 8% yield with the recovery of **1** in 80% yield. Under the electrochemical conditions, methoxide is generated at the cathode continuously and kept at low concentration, which may avoid to some extent the unwanted reduction of Co(III). But the reduction of Co(III) to Co(II) by solvent most likely occurred under the electrochemical conditions for the amination, resulting in the consumption of much more charge (usually above 5 F/mol and up to 13 F/mol, see SI for details) than the theoretical amount of 2 F/mol. This means that 5-13 equiv of Co(III) are consumed under the electrochemical conditions to exhaust the substrate. Fortunately, good yields can be obtained under electrochemical conditions because of the continuously generation of Co(III) and methoxide. These results also demonstrate the advantage of electrochemistry in promoting this type of oxidation reactions. The alkylation was conducted at room temperature and the current efficiency (2.4-2.8 F/mol) is generally higher than amination. Of course, in this case, the high efficiency of substrate oxidation by Co(III) may also contribute to the high current efficiency.

Can the electrochemical studies be repeated under heated conditions in order to see if a catalytic current could be observed?

Response: This is an excellent point. The internal temperature for the preparative electrolysis is about 72 °C. We have conducted the CV at 60 °C and have included the results as Supplementary Figure 3b. The voltammogram under heating (red) moved up a little bit from that of rt (black) probably because of the increase of the diffusion coefficient at higher temperature ($i_p \propto D^{1/2}$). We cannot go higher temperature because of the solution is no longer at stationary. The half wave potential at 60 °C also moved a bit toward negative side, suggesting faster reaction of Co(III). In short, there seems no catalytic current at 60 °C.

Supplementary Figure 3b. Cyclic voltammograms. Black: [Co]-**1** (1.5 mM), **1** (10 mM), NaOMe (1.5 mM). Red: [Co]-**1** (1.5 mM), **1** (10 mM), NaOMe (1.5 mM), tested at 60 °C.

What literature is available to support the proposal that methanol is involved in the reaction?

What is the outcome of conducting the reaction using deuterated solvent? A series of reactions should be carried out: deuterated acetonitrile, deuterated methanol, both deuterated solvent.

Response: The cathodic reduction of methanol solvent on Pt electrode to generate methoxide and H₂ is an established process (*Angew. Chem. Int. Ed.* **2019**, *58*, 3562). It is like the reduction of H₂O to generate hydroxide and H₂. Under our electrochemical conditions, MeOH is the easiest to reduce. We have previously studied the oxidation of anilides with ferrocenium in the presence of methoxide by cyclic voltammetry (*Angew. Chem. Int. Ed.* **2016**, *55*, 2226). In those studies, catalytic current can be observed to provide support for the involvement of methoxide. Here, we know from the stoichiometric studies that a base is needed for substrate oxidation. The electrochemical reaction of **1** afforded the desired product **2** in 52% yield in the absence of basic additive. These results point to the involvement of cathodic generated methoxide in the reaction to promote substrate oxidation.

We have conducted the reaction of carbamate **1** in deuterated solvents. The use of deuterated solvents did not affect the reaction efficiency. No deuterium incorporation into the product was observed. We are wondering what the reviewer hopes to know with these experiments.

Solvent	Yield of 2
MeCN/CD ₃ OD (5:1)	86%
CD ₃ CN/MeOH (5:1)	85%
CD ₃ CN/CD ₃ OD (5:1)	86%

Can the authors show that the carbamate can be deprotonated by MeO-?

Response: The pK_a of the carbamates employed in our studies are likely close to that of acetanilide (13.8, H₂O). The pK_a of MeOH is 16. The neutral carbamate substrate **1** is oxidized in MeOH/MeCN at above 1 V vs SCE (Fig 5a). In the presence of NaOMe, the substrate is oxidized at much lower potentials of about 0.3 V (Supplementary Figure 3a). This oxidation is attributed to the oxidation of the conjugate base of **1**. Note this oxidation is not for methoxide, which is oxidized at around 1 V vs SCE. These results clearly demonstrated the deprotonation of carbamate by methoxide.

Can the authors isolate or otherwise characterise any of the cobalt intermediate proposed in their catalytic cycle (for example [Co-H] or 97)?

Response: We have tried to synthesize **97** from Co(III) and carbamate **1** but have failed to do so probably because of its instability and facile reduction of Co(III) to Co(II) in the presence of a base. Co(salen)-H species have often been proposed as intermediates but have not been isolated or characterized probably due to its transient nature (*Nat. Chem.* **2020**, *12*, 747-754; *J. Am. Chem. Soc.* **2019**, *141*, 9548; *Chem. Rev.* **2016**, *116*, 8912). This type of metal-hydride has been observed by FAB-MS and FT-IR analysis (*Synthesis* **2008**, *10*, 1628-1640) but without other concrete evidence. It remains challenging to characterize this type of metal-hydride intermediates.

The experimental procedures appear to be well described.

Computational studies were completed using an inappropriate basis set. These studies should be benchmarked on either literature methods or experimental values. Coordinates for all calculated structures should be provided, or the reference to the database where they are deposited included in the supporting information.

Response: The stereoselectivity is consistent with similar radical cyclizations reported previously (ref 51). We have added single point energy corrections by M062X/def2-TZVPP. The results still support the selectivity of experimental observation. The coordinates were added in the SI.

The ¹H NMR spectra of numerous products showed that there was contamination. These products should be further purified, and the spectra re-taken. This is exemplified in products **14**, **15** and **25**.

Response: Compounds **14** and **15** were contaminated with inseparable hydroamidation products **14'** and **15'**, respectively. Other products that contained similar side products are **9** (17:1), **10** (14:1), **11** (9:1), and **12** (21:1). We have added this information to the SI. We do not see problem with **25**. C-radical elimination to give alkene and reduction to give alkane are known to be competitive reactions with Co(salen) catalysis (*J. Am. Chem. Soc.* **2019**, *141*, 9548; *J. Am. Chem. Soc.* **2014**, *136*, 16788). We have tried to figure out the pathway for the formation of the hydroamidation products by conducted the reactions with deuterated solvents. No substantial D-incorporation into the hydroamidation product was observed, suggesting that the H-atom was not from the solvent. It is likely that the intermediate cyclization-derived C-radical such as **99** in Fig 5d picked a H atom from Co-H species. Since the H atom of the Co-H intermediate comes from the substrate, there will be no D-incorporation into the hydroamidation products when the reactions are conducted in deuterated solvents. Note that this type of reduced products was not observed for the alkylation reaction.

Also note that the diastereomers (if formed) for the amination and alkylation reactions are not separable and may look like contamination. In addition, many of the alkene starting materials are mixtures of *Z/E* isomers. Because of the radical nature of the cyclization reactions, there is no need to prepare pure stereoisomers of the alkene starting materials, which is advantageous because it is sometimes nontrivial to prepare pure alkene stereoisomers. For **39** and **41**, the cyclizations are stereoselective but the starting materials were employed as diastereomeric mixtures.

Zoom ins of relevant ¹H and ¹³C NMR splitting patterns should be added. For example, the ¹H NMR region from 5.5 – 6.5 ppm of product 3.

Response: We have added zoom ins for all relevant NMR spectra.

Reviewer #2 (Remarks to the Author):

Hai-Chao Xu and coworkers have done a splendid job describing an efficient and novel means of achieving intramolecular allylic C–H functionalization. The chemistry occurs electrocatalytically using tailored cobalt-salen complexes as catalysts. The authors do an excellent job placing their work within the context of prior art and provide convincing arguments detailing the improvements their work brings to the synthesis toolbox. The substrate scope is large and the yields are excellent. The chemistry can be scaled to decagram levels and occurs without the use of external chemical oxidants. The mechanistic scheme, illustrated in Figure 5, is sound – it resting upon the results of a set of well crafted and insight providing control experiments.

I believe the research is of broad interest and will surely attract the attention of researchers in both academics and industry. The chemistry has the potential for widespread use.

I recommend publication “as is”, except for one suggestion. Thus, I believe that it would behoove the authors to include a few explicit comments regarding the manipulations that can be carried out on both the urea and the oxazolidinone containing adducts. I believe this would enhance the interest.

Response: We thank the reviewer for the very positive comments and suggestions. There are indeed known methods for the conversion of the products to valuable structures. We have added the following sentences regarding the transformations of the urea and

carbamate products: The scalability of our method coupled with versatile known transformations of the alkene-bearing cyclic carbamates and ureas products to valuable structures such as vicinal amino alcohols and diamines makes the electrocatalysis attractive for synthetic applications.^{40,46,52}

Reviewer #3 (Remarks to the Author):

Recommendation: Publish after minor revisions noted

A cobalt-salen complexes catalyzed intramolecular cyclizations enabled by electrocatalysis with a broad tolerance of functional groups and alkene substitution patterns in was described this submit. This method could be applied on a decagram scale synthesis. Control experiments and cyclic voltammetry experiments were carried out to prove the mechanism. However, it is necessary to make some revises before publication:

Response: We thank the reviewer for the very positive recommendation and suggestions.

1 Pay attention to the statement, such as the statements at Line 18 and 21; Line 144, [Co]- should be [Co]-

Response: We are confused about this suggestion. Could you specify what the problems of these statements are? [Co]⁻ on line 144 has been corrected.

2 Could it achieve enantioselectivity if using chiral Salen cobalt catalysis?

Response: This is a very good point. We have tested the following catalyst for the formation of compound **4**. The reaction afforded only 20% yield along with about 20% inseparable hydroamidation product. It is unclear at this stage if the reaction is enantioselective. It is certainly worth further exploration in the future to screen the salen ligands. One potential challenge is that the Co(III) state of the Co complex shown below is not stable as shown by its irreversible voltammogram. For this reason, catalysts **[Co]-1** and **[Co]-5** contain a bulky vicinal diamine linker that can stabilize the Co(III) state.

3 These literatures should be cited: “Acc. Chem. Res. 2019, 52, 3309–3324”; “CCS Chem. 2020, 2233-2244”; “Chin. J. Chem. 2021, 39, 143-148”.

Response: These references have been added to the manuscript as refs 37-39 .

REVIEWER COMMENTS

Reviewer #1 (Remarks to the Author):

The authors should explicitly discuss the reaction outcomes with the different catalyst loadings in the manuscript. What do the authors attribute the formation of the side product to?

The authors should explicitly discuss in the manuscript that the amination reaction worked with sub stoichiometric electrolyte.

Could the authors please explain their comment that "in some cases that the use of 1 equiv of salt was beneficial for the cyclization reaction such as the formation of carbocycles 70 and 71"?

The authors should explicitly describe in the manuscript the difference observed between the yields when different base was used. Where a range of other bases' tested? What occurred when a soluble base was tested? What are the results when using Cs₂CO₃ (as this was used in Fig 5b)?

For both amination and alkylation, can the model reaction be tested under the optimised conditions for the other reaction as a point of comparison and discussed explicitly in the text?

The reactions conducted in deuterated solvent support the authors mechanism. If the reaction involved hydrogen atom abstraction from solvent, deuterium incorporation would have been observed. This should be acknowledged in the manuscript.

Given the authors response that metal-hydrides have been observed by FAB-MS and FT-IR (also ¹H NMR should be possible), the authors should characterise the proposed [Co-H] by these methods.

The computational results are not correctly reported. The accepted convention is to report as: M06-2X/def2-TZPP//M06-2X/6-31G(d) for single point calculations. Choice of basis set and level of theory should be based on the literature. Basis set, level of theory and Gaussian16 package should all be referenced.

Explicit discussion of the hydroamination side-product should be included in the manuscript.

Reviewer #3 (Remarks to the Author):

Hai-Chao Xu and coauthors developed intramolecular allylic C-H functionalizations. I commented on the article last time and they have revised the paper carefully according to my comments. I have no further comments on the paper. So that I think the contribution is suitable to merit publication in Nature Communications.

A point-to-point response to the comments of reviewers is as following. The original reviewer comments are in black and our responses are in blue.

Reviewer #1 (Remarks to the Author):

We thank the reviewer for taking time to review the manuscript again.

The authors should explicitly discuss the reaction outcomes with the different catalyst loadings in the manuscript. What do the authors attribute the formation of the side product to?

Response: We have included in Table 1, entry 5 the results of 5 mol% of catalyst loading and added the information on the formation of the side product. The following discussion has been added to the text: Reduction of the amount of [Co]-1 to 5 mol% did not affect the yield of 2 but led to the formation of a small amount of side product 2' (4% yield) that was inseparable from 2 (entry 5).

This side product was likely formed through the reduction of the intermediate C-radical 99 with [Co-H] species or the solvent. C-radical elimination to give alkene and reduction to give alkane are known to be competitive reactions with Co(salen) catalysis (*J. Am. Chem. Soc.* **2019**, *141*, 9548; *J. Am. Chem. Soc.* **2014**, *136*, 16788). We have added the following discussion to the text: The intermediate C-radical 99 may be reduced with [Co-H] species or the solvent to generate side product 2'.

The authors should explicitly discuss in the manuscript that the amination reaction worked with sub stoichiometric electrolyte.

Response: We added the following discussion to the text: Although the reactions were conducted with 1 equiv of Et₄NPF₆, it was possible to reduce its amount to 0.1 equiv as demonstrated with the synthesis of compounds 5, 19, and 42.

Could the authors please explain their comment that “in some cases that the use of 1 equiv of salt was beneficial for the cyclization reaction such as the formation of carbocycles 70 and 71”?

Response: The yields for 70 and 71 in the presence of 1 equiv of salt were higher than in the presence of 0.1 equiv. We do not have a good explanation for this observation.

The authors should explicitly describe in the manuscript the difference observed between the yields when different base was used. Where a range of other bases' tested? What occurred when a soluble base was tested? What are the results when using Cs₂CO₃ (as this was used in Fig 5b)?

Response: A range of bases were tested as shown in Supplementary Table 1. We have added the following discussion to the text: While K_2CO_3 was similarly effective as Na_2CO_3 , other bases such as NaOPiv, $NaHCO_3$, K_2HPO_4 , and LiOMe were less efficient in promoting the formation of **2** (Supplementary Table 1). The use of Cs_2CO_3 as a base resulted in the decomposition of the carbamate **1** to generate MeOCONHPMP in 60% yield and no formation of **2** (Supplementary Table 1).

For both amination and alkylation, can the model reaction be tested under the optimised conditions for the other reaction as a point of comparison and discussed explicitly in the text?

Response: This information has already been provided in the SI. Please see Supplementary Table 1, entry 14 and Supplementary Table 2, entry 10. We have stated the following in the text "These alkylation reactions required different conditions for optimal results. After reaction optimization (Supplementary Table 2), we determined that a different cobalt-salen complex, [Co]-**5**, was necessary to achieve the best results."

The reactions conducted in deuterated solvent support the authors mechanism. If the reaction involved hydrogen atom abstraction from solvent, deuterium incorporation would have been observed. This should be acknowledged in the manuscript.

Response: The reactions are dehydrogenative reactions. Since these results do not provide additional insight into the reaction mechanism, we have included the results in Supplementary Table 4.

Given the authors response that metal-hydrides have been observed by FAB-MS and FT-IR (also 1H NMR should be possible), the authors should characterise the proposed [Co-H] by these methods.

Response: Many studies have proposed Co-H species as intermediates, but these species have not been isolated or characterized probably due to its transient nature (*Nat. Chem.* **2020**, *12*, 747-754; *J. Am. Chem. Soc.* **2019**, *141*, 9548; *Chem. Rev.* **2016**, *116*, 8912). Characterization of the Co-H species by itself would be a breakthrough in the field of cobalt catalysis and will be an independent study in the future (*J. Am. Chem. Soc.* **2015**, *137*, 4860). FAB-MS, FT-IR and NMR analysis cannot provide concrete evidence for the Co-H species. The concentration of Co-H during the electrolysis should be rather low, which makes it even more difficult to detect, especially for technologies such as for FT-IR and NMR analysis that are not that sensitive. Mass spectroscopy does not provide information on structure. Based on these considerations, we did not try to characterize the proposed Co-H species. We hope that the reviewer can give us a break on this one.

The computational results are not correctly reported. The accepted convention is to report as: M06-2X/def2-TZPP//M06-2X/6-31G(d) for single point calculations. Choice of basis set and level of theory should be based on the literature. Basis set, level of theory and Gaussian16 package should all be referenced.

Response: We have modified the description and added relevant references in the SI.

Explicit discussion of the hydroamination side-product should be included in the manuscript.

Response: We have added the side product to Table 1 as compound **2'**. The following discussion has been added to the text: Reduction of the amount of [Co]-1 to 5 mol% did not affect the yield of **2** but led to the formation of a small amount of side product **2'** (4% yield) that was inseparable from **2** (entry 5).

Reviewer #3 (Remarks to the Author):

Hai-Chao Xu and coauthors developed intramolecular allylic C–H functionalizations. I commented on the article last time and they have revised the paper carefully according to my comments. I have no further comments on the paper. So that I think the contribution is suitable to merit publication in Nature Communications.

Response: We thank the reviewer for taking time to review the manuscript again and for the positive recommendation.

REVIEWERS' COMMENTS

Reviewer #2 (Remarks to the Author):

The authors have done a thorough and excellent job responding to the reviewers. I recommend publication of this very interesting and insightful piece of work.

A point-to-point response to the comments of reviewers is as following. The original reviewer comments are in black and our responses are in blue.

Reviewer #2 (Remarks to the Author):

The authors have done a thorough and excellent job responding to the reviewers. I recommend publication of this very interesting and insightful piece of work.

Response: We thank the reviewer for taking time to review the manuscript again and for the positive recommendation.